# Genomic Aberrations Associated with the Pathophysiological Mechanisms of Neurodevelopmental Disorders

**DOI:** 10.3390/cells10092317

**Published:** 2021-09-04

**Authors:** Toshiyuki Yamamoto

**Affiliations:** Institute of Medical Genetics, Tokyo Women’s Medical University, Tokyo 162-8666, Japan; yamamoto.toshiyuki@twmu.ac.jp

**Keywords:** nonallelic homologous recombination (NAHR), contiguous gene deletion syndrome, classical microdeletion syndrome, genome disease, diagnostic yield, exome sequencing

## Abstract

Genomic studies are increasingly revealing that neurodevelopmental disorders are caused by underlying genomic alterations. Chromosomal microarray testing has been used to reliably detect minute changes in genomic copy numbers. The genes located in the aberrated regions identified in patients with neurodevelopmental disorders may be associated with the phenotypic features. In such cases, haploinsufficiency is considered to be the mechanism, when the deletion of a gene is related to neurodevelopmental delay. The loss-of-function mutation in such genes may be evaluated using next-generation sequencing. On the other hand, the patients with increased copy numbers of the genes may exhibit different clinical symptoms compared to those with loss-of-function mutation in the genes. In such cases, the additional copies of the genes are considered to have a dominant negative effect, inducing cell stress. In other cases, not the copy number changes, but mutations of the genes are responsible for causing the clinical symptoms. This can be explained by the dominant negative effects of the gene mutations. Currently, the diagnostic yield of genomic alterations using comprehensive analysis is less than 50%, indicating the existence of more subtle alterations or genomic changes in the untranslated regions. Copy-neutral inversions and insertions may be related. Hence, better analytical algorithms specialized for the detection of such alterations are required for higher diagnostic yields.

## 1. Introduction

Neurodevelopmental disorders are defined as a concept that includes a wide range of symptoms such as intellectual disability, developmental retardation, communication disorders, autism spectrum disorders, attention deficit hyperactivity disorder, learning disabilities, and motor disorders such as tics [1,2]. Cerebral palsy, epilepsy, and psychiatric disorders are also understood as peripheral diseases with the same origin. In other words, it is easy to think of the pathophysiology of many of these symptoms if we consider that some disorder of the synaptic function of the central nervous system causes various combinations of symptoms as clinical symptoms [3].

Since the completion of the Human Genome Project in 2003 (Gibbs), comprehensive genome analysis technology using the primary sequence information of the human genome has advanced, and comprehensive genome copy number analysis using microarrays and comprehensive genome analysis using next-generation sequencers have become possible. Genomic medicine using these analysis techniques has revealed the causes of neurodevelopmental disorders in children one after another [4,5]. Although the diagnostic yields in the chromosome G banding method was approximately 4%, which was the only comprehensive analysis method before the Human Genome Project, now, the diagnostic rate has increased to about 30–40% [6]. Because the genomic research of neurodevelopmental disorders is still ongoing, the involvement of the genomic alteration in neurodevelopmental disorders is not yet fully understood.

Here, the genetic factors of neurodevelopmental disorders and the current state of diagnosis by genomic medicine are outlined.

## 2. Chromosomal Deletions

Genomic copy number variations often contribute to neurodevelopmental disorders, indicating that many genes important for neurogenesis are copy-number-dependent. In general, 22q11.2 microdeletion (MIM #192430) is the most frequently observed genomic alteration, occurring in one in three-thousand live births [7]. The 22q11.2 microdeletion is caused by nonallelic homologous recombination (NAHR) facilitated by the low-copy repeats (LCRs) present at both ends of the deletions (Figure 1). Similar to the 22q11.2 microdeletion, several other microdeletions are mediated by the LCRs (Table 1). Due to these characteristics of the genome, microdeletion syndromes resulting from the adjacent LCRs are sometimes called “genome diseases”. Furthermore, the microdeletion syndromes, identified before the Human Genome Project, are characterized by prominent phenotypic features and are relatively easy to diagnose [8]. Therefore, these are called “classical microdeletion syndromes”.

For instance, patients with 22q11.2 microdeletion syndrome often present with tetralogy of Fallot, as a congenital heart disease (Table 1). Additionally, patients with Williams-Beuren syndrome (MIM #194050) and Smith-Magenis syndrome (MIM #182290) also present with congenital heart diseases associated with supraclavicular stenosis and ventricular septal defect, respectively. In addition to congenital heart diseases, patients with these syndromes exhibit distinctive features, which provide important clues for clinical diagnosis. Furthermore, the variable phenotypes in these disease groups are caused by the deletion of multiple adjacent genes, leading to the term “continuous gene deletion syndrome”.

“Classical microdeletion syndromes” or “genome diseases” are often associated with various levels of neurodevelopmental abnormalities, because the deleted region contains genes related to neurodevelopment, which are copy-number-dependent. Furthermore, a condition where heterozygous deletions or loss of homologous alleles occur and the remaining functional copy of the gene is incapable of producing a sufficient gene product required for maintaining the normal function is referred to as haploinsufficiency. Haploinsufficiency of genes related to neurodevelopment is an essential mechanism in classical microdeletion syndromes.

## 3. Microduplications

The genes that cause neurodevelopmental delay upon their deletion are often copy-number-dependent. These genes affect the neurodevelopmental process, not only by deletions, but also by duplications, (e.g., reciprocally increased number of gene copies, such as in chromosomal partial trisomy). In fact, it is known that neurodevelopmental disorders, such as autism and attention deficit hyperactivity disorder, occur when the regions responsible for classical chromosomal microdeletion syndrome are duplicated (Table 1). Smith-Magenis syndrome is caused by the deletion of the 17p11 region [9], whereas, reciprocal duplication of this region causes Potocki-Lupski syndrome (MIM #610883), and the patients present with relatively severe developmental disorders [10]. The *RAI1* gene, located on 17p11, is considered to be responsible for the neurodevelopmental disability in both Smith-Magenis and Potocki-Lupski syndromes [11]. In addition, developmental disorders occur when the regions responsible for 22q11.2 microdeletion and Williams syndromes are duplicated.

## 4. Different Symptoms Are Associated with Deletion and Duplication of Certain Genes

Several genes are known to show different clinical symptoms depending on their deletion or duplication (Table 2). For instance, deletion of *PMP22* (located on 17p12) causes hereditary neuropathy with susceptibility to pressure palsies (MIM #162500) [12], while its duplication causes Charcot-Marie-Tooth disease (MIM #118220) [13]. Similarly, deletion of *PLP1* (located on Xq22.2) causes spastic paraplegia associated with peripheral neuropathy; however, its duplication causes a congenital white matter abnormality, known as Pelizaeus-Merzbacher disease (MIM #312080) [14,15]. These differences can be attributed to different mechanisms associated with gene deletion or duplication events [16]. Additionally, it is believed that duplication events result in the increased expression of genes, inducing cell stress.

Furthermore, mutation of *MECP2* gene (located on Xq28) causes Rett syndrome (MIM #312750), a neurodevelopmental disorder specific to females; however, its duplication (MIM #300815) is asymptomatic in women, while causing severe intellectual disability, epilepsy, and susceptibility to infection in males [17]. The exact mechanism underlying *MECP2* deletion or duplication is unclear to date [18].

## 5. Significance of Microarray in Detecting Chromosomal Aberrations

Since 2010, chromosomal microarray testing has been commonly used for detecting chromosomal aberrations, and it has helped in the diagnosis of several previously unknown chromosomal microdeletion syndromes [19,20]. Among these, a few are novel genomic diseases that are caused by LCR-mediated NAHR; one such disease is 16p11.2 microdeletion syndrome (MIM #611913) [21]. The 16p11.2 microdeletion is relatively frequent and is observed in approximately 1/100 patients with autism. Furthermore, deletion or duplication of 16p11.2 causes similar developmental disorders, and their clinical diagnosis is difficult, contrary to the classical microdeletion syndromes, as the patients have very few differentiating symptoms. Hence, comprehensive copy number variation (CNV) analysis by microarray is the only diagnostic method for 16p11.2 microdeletion syndrome.

The chromosomal microdeletions caused by LCR-mediated NAHR are limited, and various chromosomal aberrations detected by microarray are caused by random breakpoints (Table 3). However, even if the breakpoints are not common, the chromosomal microdeletions that can be clinically classified as the same entities due to the common clinical symptoms include the main gene(s) in the deleted region (Figure 2). Thus, the clinical symptoms can be diagnosed because of the involvement of the main gene(s) in the deleted regions.

## 6. Genes Identified Based on Their Genomic Copy Number Changes

In 2011, we identified a small deletion in Xq11.1, in a patient with epileptic encephalopathy [22]. The deleted region contained the *ARHGEF9* gene. Additionally, we identified a nonsense mutation in *ARHGEF9* in a different patient with epileptic encephalopathy. Based on these findings, *ARHGEF9* has been registered as the causative gene for developmental and epileptic encephalopathy 8 (MIM #300607) [23].

Further, in 2011, we reported two cases of 5q31 microdeletion for the first time [24]. Both patients exhibited common clinical symptoms with infantile epileptic encephalopathy and shared severe psychomotor development. Following our study, two other studies reporting overlapping chromosomal microdeletions narrowed down the candidate gene responsible for the syndrome to be *PURA* [25,26]. Finally, next-generation sequencing (NGS) of patients with severe psychomotor development and infantile epileptic encephalopathy revealed a large number of de novo mutations in *PURA*, confirming the association of *PURA* with 5q31 microdeletion syndrome [27]. Hence, currently, 5q31 microdeletion syndrome is known as a *PURA*-related neurodevelopmental disorder.

In another study, we found a 15q14 microdeletion in a patient with mild neurodevelopmental disorder with ventricular septal defect and submucosal cleft palate [28]. Further, the deleted region contained *MEIS2*, which has since been identified as the causative gene for neurodevelopmental disorders associated with cleft palate and congenital heart disease [29].

Hence, as discussed above, when the phenotype caused by chromosomal deletion and gene mutation is the same, it is considered to be caused by haploinsufficiency and is relatively easy to understand.

## 7. Genes Whose Phenotypes Are Not Affected by Genomic Copy Number Changes

*ZBTB20*, located at 3q13.31, has been identified as the causative gene for Primrose syndrome (MIM #259050), which is associated with severe neurodevelopmental disorders [30]. Previously, we found that the symptoms associated with neurodevelopmental disorders were very mild and inconsistent in the cases with 3q13 deletion compared to those observed in Primrose syndrome [31]. Therefore, Primrose syndrome is unlikely to be caused by haploinsufficiency of *ZBTB20* and is thought to be the result of the dominant negative effect of *ZBTB20* mutations.

*SATB2* is located at 2q33.1 and is known as the causative gene for Glass syndrome (MIM #612313), which causes characteristic symptoms, such as intellectual disability and dentition malformation. Patients with *SATB2* mutations and deletions show similar symptoms [32,33]. Furthermore, *HECW2* is located on the 3-Mb centromeric side of *SATB2* at 2q32.3-q33.1 and has recently been identified as a causative gene for neurodevelopmental disorders with hypotonia, seizures, and absent language (NDHSAL; MIM #617268) [34,35]. However, microdeletion of 2q32.3-q33.1 is not known to cause severe developmental disorders. Hence, the neurodevelopmental disorders due to *HECW2* mutations are considered to be because of the dominant negative effect [36].

Thus, the pathomechanism of neurodevelopmental disorders can be revealed by understanding whether the gene mutation is due to haploinsufficiency or the dominant negative effect. Therefore, it is important to compare the phenotypes of patients due to gene deletions and the gene mutations associated with the dominant negative effects.

## 8. Diagnostic Yield of the Methods Used for Genetic Testing

The diagnostic yield of chromosomal microarray testing has been found to be 12–20% [20,37,38]. Furthermore, the diagnostic yield does not vary depending on the type of platform used, such as comparative genomic hybridization or single-nucleotide polymorphism, indicating that it does not depend on the resolution of the microarrays.

When there are no pathogenic CNVs, single-nucleotide variants (SNVs) may be associated with the occurrence of the diseases. Currently, NGS-based exome sequencing is recommended to detect SNVs [1]. The diagnostic yield of exome sequencing is approximately 30% [6]. Thus, more than 40% of the cases can be diagnosed using either genome copy number analysis or exome sequencing. However, for the remaining patients (more than 50%), the genomic background of the diseases remains unclear (Figure 3).

## 9. Undetected Genomic Backgrounds

What are the causes of the diseases where no pathogenic CNVs or SNVs have been observed? One possible cause in such cases is the genomic copy number aberrations; however, they are very small such that cannot be detected by microarray with the general resolution. For instance, most patients with Duchenne muscular dystrophy (DMD; MIM #310200, 71%) show exonic deletions in the *DMD* gene, and only 17% of the patients show pathogenic SNVs [39]. As this is a well-known phenomenon, multiplex ligation probe amplification is primarily used for the diagnosis of *DMD*. Previously, we identified an exonic deletion in *MED13L*. The microarray results showed an aberrant log2 ratio in only three probes; hence, we used a different method to confirm the deletion [40]. Thus, such small CNVs could be misdiagnosed by microarray and exome sequencing.

The other possible mechanisms are silent mutations, deep intronic variants, aberrations in the noncoding regions, and genomic methylation. A few silent mutations and deep intronic variants are known to affect the splicing machinery [41,42,43]. As these variants are generally excluded during the filtering of exome sequencing data, those affecting the splicing machinery may have been overlooked. Furthermore, aberrations in the noncoding regions cannot be detected by exome sequencing, and scientific evidence of their association with disease occurrence is insufficient. Additionally, altered methylation of the wild-type sequences is known to be the underlying cause of a few diseases. However, methylation abnormalities cannot be detected by microarray and exome sequencing.

However, even if such abnormalities are detected, it may be difficult to confirm their association with the diseases. For instance, the precise detection of small CNVs is not possible. However, while targeting small CNVs, a large number of nonpathogenic CNVs may be detected, making it difficult to distinguish the pathogenic CNVs. A similar problem may arise with other discussed mechanisms. Exome sequencing covers only approximately 2% of the entire genome. However, sequencing the entire genome may result in a large number of variants, making the visual filtering difficult. Thus, analyzing the entire genome is not practical unless a more accurate database is developed and automated filtering using artificial intelligence is introduced.

## 10. Novel Developments Expected in Whole Genome Analysis for the Detection of Chromosomal Aberrations

To date, we have analyzed CNVs in many patients. Among the analyzed cases, a few have exhibited comparatively more complicated structural abnormalities, such as three consecutive deletions and additional triplications in the duplicated fragment [44,45]. Further, we performed whole-genome analysis to clarify the patterns of structural abnormalities [46]. The results suggested that seemingly simple structural abnormalities may be caused by more complex changes, such as inversions and insertions.

Thus, there is a possibility that the copy-neutral rearrangements, such as inversions or insertions, contribute to disease occurrence. However, they cannot be detected through microarray (Figure 4), as evidenced in the literature [47,48,49]. Hence, with the usage of whole-genome analysis and the availability of appropriate algorithms or analysis software that can efficiently detect inversions and insertions without copy number changes, the diagnostic yields of the disease-causing genomic backgrounds will increase.

## 11. Conclusions

The involvement of genomic alterations in neurodevelopmental disorders and the progress of their analysis technology were outlined. Chromosomal microarray testing is positioned as the first-tier testing for undiagnosed neurodevelopmental disorders [20]. In approximately 15% of patients, pathogenic CNVs are expected to be detected, and the final diagnosis would be obtained. If the diagnosis cannot be obtained by the chromosomal microarray testing, SNVs are recommended to be comprehensively analyzed by exome analysis. Exome analysis will reveal pathogenic SNVs in approximately 30% of patients. If neither microarray chromosomal testing nor exome analysis show pathogenic variants, there may be variants that cannot be detected by these techniques. Exonic deletions/duplications, copy-neutral inversions/insertions, and variants in noncoding regions would have been underdiagnosed. Therefore, it is expected that whole-genome analysis will detect such alterations that have not been found so far and that the diagnostic yield will be further improved (Figure 5). The genomic basis of neurodevelopmental disorders has not yet been fully elucidated, and genomic testing methods will be refined further in the future.

## Figures and Tables

**Figure 1 cells-10-02317-f001:**
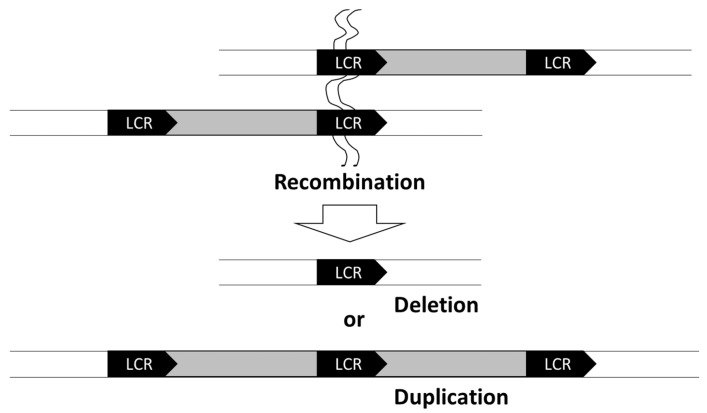
Schematic representation of the mechanism of nonallelic homologous recombination. Deletions and duplications of the regions of interest (grey rectangles) can be caused by nonallelic homologous recombination triggered by the presence of low-copy repeats (LCRs).

**Figure 2 cells-10-02317-f002:**
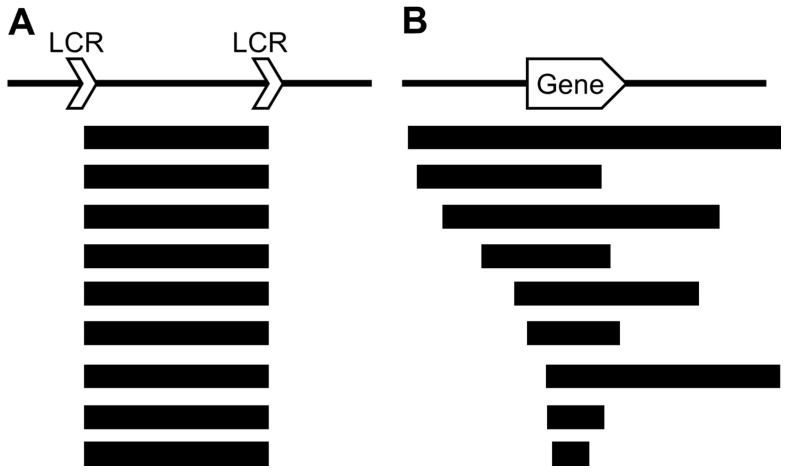
Schematic representation of the patterns of deletions. (**A**) Deletions (black rectangles) caused by nonallelic homologous recombination triggered by surrounding LCRs show the same breakpoints in patients. (**B**) Deletions (black rectangles) with random breakpoints, however, include specific gene(s).

**Figure 3 cells-10-02317-f003:**
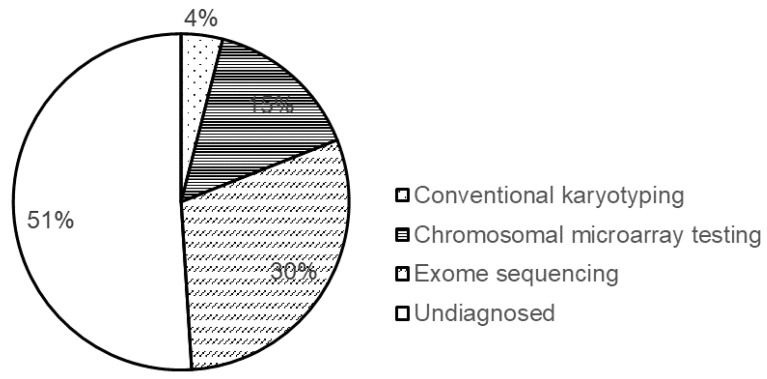
Schematic representation of diagnostic yields of various methods used for genetic testing. Diagnostic yields of the methods are shown in % plot format.

**Figure 4 cells-10-02317-f004:**
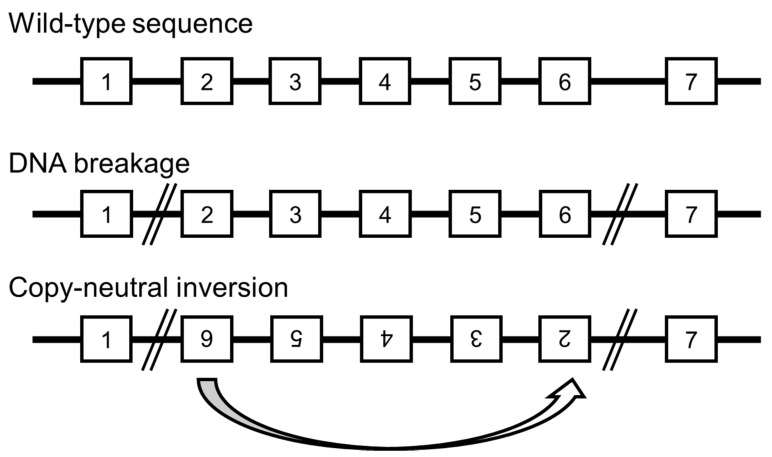
Possible mechanism of copy-neutral inversion that cannot be detected by chromosomal microarray testing and exome sequencing.

**Figure 5 cells-10-02317-f005:**
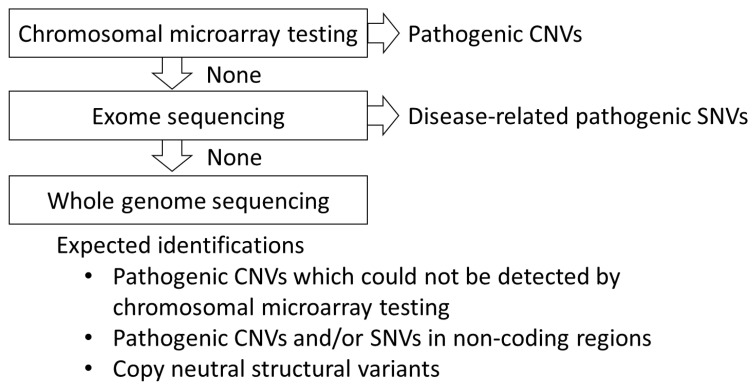
Schematic representation of the stepwise diagnosis for patients with neurodevelopmental disorder.

**Table 1 cells-10-02317-t001:** Classical microdeletion syndromes and reciprocal duplications.

Chromosomal Regions	Deletions		Duplications	
	Microdeletion Syndromes	Main Clinical Features	Microduplication Syndromes	Main Clinical Features
22q11.2	22q11.2 deletion syndrome	Tetralogy of Fallot, language delay, distinctive facial features	22q11.2 duplication syndrome	ADHD
7q11.23	Williams-Beuren syndrome	Supraventricular stenosis, intellectual disability, distinctive facial features	7q11.23 duplication syndrome	Speech delay and autism spectrum behaviors
15q11	Prader-Willi syndrome	Developmental delay, hypotonia, obesity	15q11 duplication	Intellectual disability, autism spectrum behaviors
	Angelman syndrome	Developmental delay, epilepsy, distinctive facial features		
17p11	Smith-Magenis syndrome	Congenital heart defects, developmental delay, distinctive facial features	Potocki-Lupski syndrome	Intellectual disability, autism spectrum behaviors
5q35	Sotos syndrome	Developmental delay, macrocephaly	5q35 duplication	

**Table 2 cells-10-02317-t002:** The genes with different phenotypes in deletions and duplications.

	Deletion	Duplication
*PMP22*	hereditary neuropathy with susceptibility to pressure palsies (HNPP)	Charcot-Marie-Tooth disease
*PLP1*	spastic paraplegia	Pelizaeus-Merzbacher disease
*MECP2*	Rett syndrome in female	MECP2 duplication syndrome in male

**Table 3 cells-10-02317-t003:** Chromosomal regions and phenotypes.

	Regions	Responsible Gene(s)	Phenotypes
Microdeletions/duplications derived from NAHR			
	1q21.1 deletion/duplication		Developmental delay, distinctive features, congenital anomalies
	3q29 deletion	*DLG1*, *PAK2*	Developmental delay, psychiatric symptoms
	15q13.3 deletion	*CHRNA7*	Intellectual disability, epilepsy
	16p11.2 deletion/duplication		Developmental disorder
	17q12 deletion/duplication	*HNF1B*	Maturity onset diabetes of the young (MODY)
	17q21.31 deletion/duplication	*CRHR1, MAPT*	Developmental delay, muscular hypotonia, distinctive features
Microdeletions/duplications derived from random breakpoints			
	1q32 deletion	*IRF6*	Van der Woude syndrome
	1q41q42 deletion	*DISP1*	Developmental delay, epilepsy, distinctive features
	2p15-p16.1 deletion		Autism spectrum disorder
	2q23.1 deletion	*MBD5*	Severe developmental delay, epilepsy, microcephaly
	2q33 deletion/duplication	*SATB2*	Intellectual disability
	3p21.31 deletion	*BSN*	Developmental delay, white matter abnormality, hyperCKemia
	3q13.31 deletion	*ZBTB20*	Language delay
	5q14 deletion	*MEF2C*	Severe developmental delay, epilepsy, brain abnormalities
	5q31.3deletion	*PURA*, *NRG2*	Severe developmental delay, epilepsy
	8q24 deletion	*EXT1*, *TRPS1*	Langer-Giedion syndrome
	9q22.3 deletion	*PTCH1*	Gorlin syndrome
	10q22 deletion	*KAT6B*	Ohdo syndrome
	10q23 deletion	*PTEN*	Juvenile polyposis
	11p13 deletion	*WT1*, *PAX6*	WAGR syndrome
	11p11.2 deletion	*EXT2*, *ALX4*	Potocki-Shaffer syndrome
	12q24.21 deletion	*MED13L*	Intellectual disability
	13q32 deletion	*ZIC2*	Holoprosencephaly
	15q22.2 deletion	*NRG2*, *RORA*	Developmental delay, epilepsy
	16q24.3 deletion	*ANKRD11*, *ZNF778*	Autism spectrum disorder
	17p13.1 deletion	*DLG4*, *GABARAP*	Intellectual disability, epilepsy
	18q12.3 deletion	*SETBP1*	Language delay
	18q21.2 deletion	*TCF4*	Pitt-Hopkins syndrome
	19p13.2 deletion	*NFIX*	Malan syndrome
	Xp22.3 deletion	*KAL1*	Kallmann syndrome
	Xp21-22 deletion	*CDKL5*, *ARX*	Epileptic encephalopathy
	Xp11.4 deletion	*CASK*	Developmental delay, microcephaly
	Xp11.22 deletion	*HUWE1*	Developmental delay
	Xq11.1 deletion	*ARHGEF9*	Developmental delay, epilepsy
	Xq28 duplication	*MECP2*	Developmental delay, epilepsy

## Data Availability

Not applicable.

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
