# Peer review of "Genomic Aberrations Associated with the Pathophysiological Mechanisms of Neurodevelopmental Disorders"

_cells, 2021, doi:10.3390/cells10092317_

Round 1

Reviewer 1 Report

Excellent piece of work by the author. The manuscript reviews on the genomic underpinnings and clinical/diagnostic methods employed for neurodevelopmental disorders. Herein, author has elegantly discussed the updates in the field and have pointed out the key gaps in knowledge. Here are few very important comments for the author to consider in the revision-

1) Please explain with examples/citation in the concluding section on- what better analytical algorithms can be implemented or used for early and accurate detection of genomic anomalies and instabilities in neurodevelopmental disorders. 

2) The figure number 2 needs to be improved. Please explain what those black bars mean and represent. Explain in the legends section. Provide an example of a disease model to explain and help readers understand the mechanistic basis.

3) Figure 3- The pie chart must be clarified as % plot format in the legends section.

4) Please clarify the Japanese characters in Table 3, Page 5.

5) In figure 4, author has explained on copy-neutral inversion phenomenon and how this fails to be detected by existing methods. Please explain with examples what methods can detect or how innovation in molecular methods can be done to capture these phenomenon.

The draft needs a conclusion section where the anticipated innovations and approaches can be discussed. 

Author Response

Thank you very much for your careful review. The provided comments were valuable for me. According to the suggestions, those were revised.

1) Please explain with examples/citation in the concluding section on- what better analytical algorithms can be implemented or used for early and accurate detection of genomic anomalies and instabilities in neurodevelopmental disorders.

> According to the suggestion, the new section (Conclusion) was added.

2) The figure number 2 needs to be improved. Please explain what those black bars mean and represent. Explain in the legends section. Provide an example of a disease model to explain and help readers understand the mechanistic basis.

> According to the suggestion, the figure legends were revised.

3) Figure 3- The pie chart must be clarified as % plot format in the legends section.

> According to the suggestion, the figure was revised.

4) Please clarify the Japanese characters in Table 3, Page 5.

> According to the suggestion, those were revised.

5) In figure 4, author has explained on copy-neutral inversion phenomenon and how this fails to be detected by existing methods. Please explain with examples what methods can detect or how innovation in molecular methods can be done to capture these phenomenon.

> It is already mentioned in the main text as “with the usage of whole-genome analysis and the availability of appropriate algorithm or analysis software that can efficiently detect inversions and insertions without copy number changes, the diagnostic yields of the disease-causing genomic backgrounds will in-crease.”

The draft needs a conclusion section where the anticipated innovations and approaches can be discussed.

> According to the suggestion, the new section (Conclusion) was added and the technologies and the methods available to identify genomic aberrations were discussed again.

Reviewer 2 Report

Review manuscript described by Toshiyuki Yamamoto “Genomic Aberrations Associated with the Pathophysiological Mechanisms of Neurodevelopmental Disorders” is very interesting to me and significant especially in relation to etiology of Neurodevelopmental Disorders. A number of review articles are published on this topic, and to upgrade the quality of this manuscript I would suggest following:

  1. The introduction of neurodevelopment disorders in respect to genomic aberration need to elaborate more.
  2. It would be nice if introduction section would have incidence of neurological diseases in relation to genomic aberration that leads to social and financial burden.
  3. The different type of genomic abnormalities needs to support with molecular pathology in relation to particular disease to make this manuscript best fit for this journal. What group (age, gender or any other group) is more susceptible to specific disease.
  4. At last, the number of technologies and methods available to identify these aberrations need to explain and details.

Author Response

Thank you very much for your careful review. The provided comments were valuable for me. According to the suggestions, those were revised.

  1. The introduction of neurodevelopment disorders in respect to genomic aberration need to elaborate more.

> According to the suggestion, the new section (Introduction) was added.

  1. It would be nice if introduction section would have incidence of neurological diseases in relation to genomic aberration that leads to social and financial burden.

> It is too early to discuss the economic effects, so the discussion for social and financial burden could not be included in the introduction section.

  1. The different type of genomic abnormalities needs to support with molecular pathology in relation to particular disease to make this manuscript best fit for this journal. What group (age, gender or any other group) is more susceptible to specific disease.

> As mentioned in the main text, the genomic basis of neurodevelopmental disorders has not yet been fully elucidated and genomic research for neurodevelopmental disorder is still on the midway. Thus, it would be inappropriate to discuss molecular pathology in relation to particular disease at present.

  1. At last, the number of technologies and methods available to identify these aberrations need to explain and details.

> According to the suggestion, the new section (Conclusion) was added and the technologies and the methods available to identify genomic aberrations were discussed again.

Round 2

Reviewer 2 Report

The revised manuscript is modified accordingly and now it is looking in good shape for readers. It could be accepted for the publication.